# ARE WIDER NETS BETTER GIVEN THE SAME NUMBER OF PARAMETERS?

**Anna Golubeva**[*]
Perimeter Institute for Theoretical Physics
Waterloo, Canada
agolubeva@pitp.ca

**Behnam Neyshabur**
Blueshift, Alphabet
Mountain View, CA
neyshabur@google.com

**Guy Gur-Ari**
Blueshift, Alphabet
Mountain View, CA
guyga@google.com

## ABSTRACT

Empirical studies demonstrate that the performance of neural networks improves with increasing number of parameters. In most of these studies, the number of parameters is increased by increasing the network width. This begs the question: Is the observed improvement due to the larger number of parameters, or is it due to the larger width itself? We compare different ways of increasing model width while keeping the number of parameters constant. We show that for models initialized with a random, static sparsity pattern in the weight tensors, network width is the determining factor for good performance, while the number of weights is secondary, as long as the model achieves high training accuarcy. As a step towards understanding this effect, we analyze these models in the framework of Gaussian Process kernels. We find that the distance between the sparse finite-width model kernel and the infinite-width kernel at initialization is indicative of model performance.[1]

## 1 INTRODUCTION

Deep neural networks have shown great empirical success in solving a variety of tasks across different application domains. One of the prominent empirical observations about neural nets is that increasing the number of parameters leads to improved performance (Neyshabur et al., 2015; 2019; Hestness et al., 2017; Kaplan et al., 2020). The consequences of this effect for model optimization and generalization have been explored extensively. In the vast majority of these studies, both empirical and theoretical, the number of parameters is increased by increasing the width of the network (Neyshabur et al., 2019; Du et al., 2019; Allen-Zhu et al., 2019). Network width itself on the other hand has been the subject of interest in studies analyzing its effect on the dynamics of neural network optimization, e.g. using Neural Tangent Kernels (Jacot et al., 2018; Arora et al., 2019) and

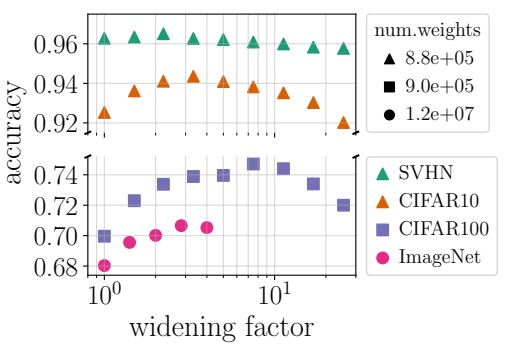

Figure 1: Test accuracy of ResNet-18 as a function of width. Performance improves as width is increased, even though **the number of weights is fixed**. Please see Section 2.3 for more details.

---

[*]Work done while an intern at Blueshift.
[1]Code is available at https://github.com/google-research/wide-sparse-nets

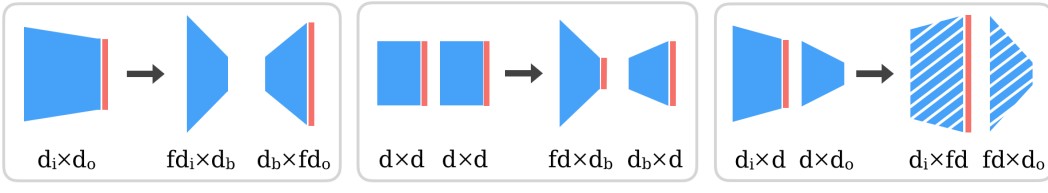

| (a) Linear Bottleneck | (b) Non-linear Bottleneck | (c) Static Sparsity |

Figure 2: Schematic illustration of the methods we use to increase network width while keeping the number of weights constant. Blue polygons represent weight tensors, red stripes represent non-linear activations, and diagonal white stripes denote a sparsified weight tensor. We use f to denote the widening factor.

Gaussian Process Kernels (Wilson et al., 2016; Lee et al., 2017).

All studies we know of suffer from the same fundamental issue: When increasing the width, the number of parameters is being increased as well, and therefore it is not possible to separate the effect of increasing width from the effect of increasing number of parameters. *How does each of these factors — width and number of parameters — contribute to the improvement in performance?* We conduct a principled study addressing this question, proposing and testing methods of increasing network width while keeping the number of parameters constant. Suprisingly, we find scenarios under which most of the performance benefits come from increasing the width.

## 1.1 OUR CONTRIBUTIONS

In this paper we make the following contributions:

- We propose three candidate methods (illustrated in Figure 2) for increasing network width while keeping the number of parameters constant.
  - (a) **Linear bottleneck**: Substituting each weight matrix by a product of two weight matrices. This corresponds to limiting the rank of the weight matrix.
  - (b) **Non-linear bottleneck**: Narrowing every other layer and widening the rest.
  - (c) **Static sparsity**: Setting some weights to zero using a mask that is randomly chosen at initialization and remains static throughout training.
- **We show that performance can be improved by increasing the width, without increasing the number of model parameters.** We find that test accuracy can be improved using method (a) or (c), while method (b) only degrades performance. However, we find that (a) suffers from another degradation caused by the reparameterization, even before widening the network. Consequently, we focus on the sparsity method (c), as it leads to the best results and is applicable to any network type.
- We empirically investigate different ways in which random, static sparsity can be distributed among layers of the network and, based on our findings, propose an algorithm to do this effectively (Section 2.3).
- We demonstrate that the improvement due to widening (while keeping the number of parameters fixed) holds across standard image datasets and models. Surprisingly, we obesrve that for ImageNet, increasing the width according to (c) leads to almost identical performance as when we allow the number of weights to increase along with the width (Section 2.3).
- To understand the observed effect theoretically, we study a simplified model and show that the improved performance of a wider, sparse network is correlated with a reduced distance between its Gaussian Process kernel and that of an infinitely wide network. We propose that reduced kernel distance may explain the observed effect (Section 3).

## 1.2 RELATED WORK

Our work is similar in nature to the body of work studying the role of overparametrization and width. Neyshabur et al. (2015) observed that increasing the number of hidden units beyond what

is necessary to fit the training data leads to improved test performance, and attributed this to the inductive bias of the optimization algorithm. Soltanolkotabi et al. (2018); Neyshabur et al. (2019); Allen-Zhu et al. (2019) further studied the role of over-parameterization in improving optimization and generalization. Woodworth et al. (2020) studied the implicit regularization of gradient descent in the over-parameterized setting, and Belkin et al. (2019) investigated the behavior at interpolation. Furthermore, Lee et al. (2017) showed that networks at initialization become Gaussian Processes in the large width limit, and Jacot et al. (2018) showed that infinitely wide networks behave as linear models when trained using gradient flow. Lee et al. (2020) systematically compared these different theoretical approaches. In all the above works, the number of parameters is increased by increasing the width. However, in this work, we conduct a controlled study of the effect of width by keeping the number of parameters fixed.

Perhaps Littwin et al. (2020) is the closest work to ours, which investigates overparametrization achieved through ensembling as opposed to layer width. Ensembling is achieved by connecting several networks in parallel into one "collegial" ensemble. They show that for large ensembles the optimization dynamics simplify and resemble the dynamics of wide models, yet scale much more favorably in terms of number of parameters. However, the method employed there is borrowed from the ResNeXt architecture (Xie et al., 2016), which involves altering the overall structure of the network as width is increased. In this work we try to make minimal changes to network structure, in order to isolate the effect of width on network performance.

Finally, static sparsity is the basis of the main method we use to increase network width while keeping the number of parameters fixed. There is a large body of work on the topic of sparse neural networks (Park et al., 2016; Narang et al., 2017; Bellec et al., 2018; Frankle and Carbin, 2018; Elsen et al., 2019; Gale et al., 2019; You et al., 2019), and many studies derive sophisticated approaches to optimize the sparsity pattern (Lee et al., 2018; Evci et al., 2019; Wang et al., 2020; Tanaka et al., 2020). In our study, however, sparsity itself is not the subject of interest. In order to minimize its effect in our controlled experiments, the sparsity pattern that we apply is randomly chosen and static. A recent work (Frankle et al., 2020) demonstrates that a random, fixed sparsity pattern leads to equal performance as the more involved methods for pruning applied at initialization, when the per-layer sparsity distribution is the same. However, we do not explore this direction in our study.

## 2 EMPIRICAL INVESTIGATION

In this section, we first explain our experimental methodology and then investigate the effectiveness of different approaches to increase width while keeping the number of parmeters fixed. Finally, we discuss the respective roles of width and the number of parameters in improving performance.

### 2.1 METHODOLOGY

In order to identify how width and number of parameters separately affect performance, we need to decouple these quantities. For a fully-connected layer, width is the number of hidden units, and for a convolutional layer, width corresponds to the number of output channels. Increasing the width of one layer ordinarily entails an increase in the number of weights. Therefore, some adjustment is required to keep the total number of weights constant. This adjustment can be made at the level of layers by reducing some other dimension (i.e., by introducing a "bottleneck"), or at the level of weights by setting some of them to be zero. When choosing a method for our analysis, we take particular care about possible confounding variables, as many aspects of a neural network are intertwined. For example, we prefer to keep the number of non-linear layers in the network fixed, as it has been shown that changing it can significantly affect the experssive power, optimization dynamics and generalizaiton properties of the network. With these constraints in mind, we specify three different methods listed in Section 1.1. In this section, we discuss these methods in detail, evaluate them, and present the results obtained on image classification tasks.

In summary, our approach is as follows: We select a network architecture which specifies layer types and arrangement (e.g., MLP with some number of hidden layers, or ResNet-18), set layer widths, and count the number of weights. We refer to this model as the *baseline*, and derive other models from it by increasing the width while keeping the number of weights constant. In this way, we obtain a

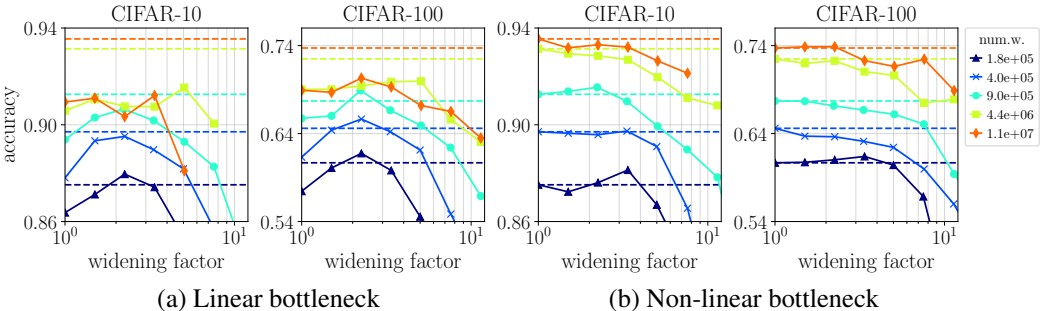

Figure 3: Best test accuracy obtained by ResNet-18 models widened using the bottleneck methods. The performance of the baseline network (before introducing bottleneck layers) is denoted by dashed lines. The legend indicates the number of weights in the baseline network. In (a), it is equal to the number of weights before introducing bottleneck layers. In (b), it is equal to the total number of weights.

family of models that we train using the same training procedure, and compare their test accuracies. For comparison and as a sanity check, we also consider the dense variants of the wider models.

We tested a variety of simple models, and observed the same general behavior in the context of our research question. For the discussion in this paper, we focus on two model types: a MLP with one hidden layer, and a ResNet-18. We chose the ResNet-18 architecture because it is a standard model widely used in practice. Its size is small enough that it allows us to increase the width substantially, yet it has sufficient representational power to obtain a nontrivial accuracy on standard image datasets. When creating a family of ResNet-18 models, we refer to the number of output channels of the first convolutional layer as the *width*, and do not alter the default width ratios of all following layers. Further details of each experiment and figure are specified in Appendix A.

## 2.2 BOTTLENECK METHODS

Here we discuss the two bottleneck methods described in the previous section and in Figure 2.

**Linear Bottleneck:** Substitute each weight matrix $W \in \mathbb{R}^{d_i \times d_o}$ by $W_1 W_2$, where $W_1 \in \mathbb{R}^{d_i \times d_b}$, $W_2 \in \mathbb{R}^{d_b \times d_o}$ and $d_b \leq \min(d_i, d_o)$.[2] Note that if $d_b = \min(d_i, d_o)$, then this reparametrization has the exact same expressive power [3] as the original network. Now, one can make the network wider by increasing the number of hidden units ($d_i$ and $d_o$ here) and reducing the bottleneck size $d_b$. This would correspond to imposing a low-rank constraint on the original weight matrix $W$. Linear bottleneck has been proposed as a way to reduce the number of parameters and improve training speed (Ba and Caruana, 2014; Urban et al., 2016). The caveat of this method is that, even though the expressive power of the reparametrized network is the same as the original one, the reparameterization changes the gradient descent trajectory and hence can affect the final results. It is therefore not possible to control the implicit regularization caused by this transformation. Note that substituting the weight matrix by a product of two matrices changes the number of parameters of the model.

**Non-linear Bottleneck:** One way to create a non-linear bottleneck is to split each layer in two as described above and add a non-linearity to the first layer. The problem with this approach is that adding a non-linearity changes the expressive power of the model, and hence adds a factor that we cannot control. An alternative approach, which is more favorable in our case and which we adopt here, is to modify the layers in pairs. The input dimension of the first layer is increased, while its output dimension (and consequently the input dimension of the second layer) is reduced ($d_b < d$). Non-linear bottlenecks are widely used in practice, particularly as a way to save parameters in very deep architectures, such as deeper variants of ResNet (He et al., 2016) and DenseNet (Huang et al., 2017).

The performance of these methods on CIFAR-10 and CIFAR-100 datasets is demonstrated in Figure 3. The results indicate that increasing width using the linear bottleneck indeed leads to improved

---

[2]In the case of a ResNet-18 model, each convolutional layer is replaced by two convolutional layers with kernel dimensions $3 \times 3$ and $1 \times 1$, respectively, and with $d_b = d_i = d_o$ before widening.

[3]Here, expressive power refers to the set of functions that can be represented by the network.

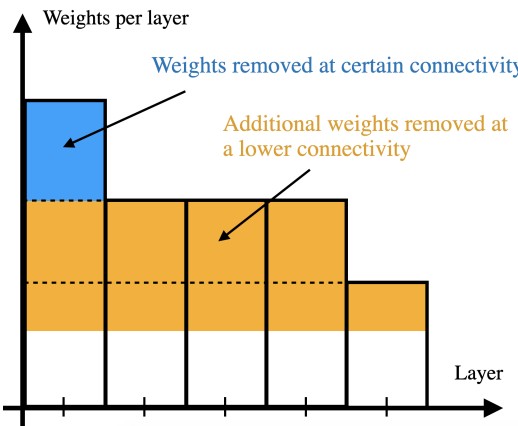

Figure 4: Illustration of our algorithm for distributing sparsity over model layers. The code implementing this algorithm is included in Appendix B.

accuracy up to a certain width. Moreover, this effect is more pronounced in smaller models that are less overparameterized. This is similar to the improvement gained when increasing the width along with the number of parameters (without a bottleneck): The improvement tends to be diminished when the base network is wider. However, as discussed above, the act of substituting a weight matrix by a product of two weight matrices changes the optimization trajectory which could in turn affect generalization. Indeed, the performance of the original model, indicated by dashed horizontal lines in the Figure, is significantly higher than that of the transformed models. Therefore, even though the width increase at a constant number of weights with the linear bottleneck method improves the result of the most narrow model obtained after the transformation, it typically does not outperform the default ResNet-18 model. Due to lack of control over the effect of inductive bias, we conclude that this choice does not qualify to be used in our controlled experiments.

The non-linear bottleneck method does not suffer from the same issues as the linear version in terms of inductive bias of reparametrization. However, as Figure 3 demonstrates, no empirical improvement is obtained by increasing the width, except for the model with $1.8e{+}05$ weights (and then the improvement is mild). Therefore, we conclude that the non-linear bottleneck model does not show a significant enough improvement to be considered as an effective method for our study.

## 2.3 SPARSITY METHOD

We now turn to the sparsity method illustrated in Figure 2c, which is the main method considered in this work. We start with a baseline model that has dense weight tensors. We then increase the width by a widening factor $f$ and sparsify the weight tensors such that the total number of trainable weights is the same as in the baseline model. In an attempt to run a controlled experiment and minimize the differences between the sparsified setup and the baseline setup, we choose the sparsity mask at random at initialization and keep it static during training. In this sense, our method differs from most other pruning methods discussed in the literature, where the aim is to maximize performance. The advantage of the sparsity method over the bottleneck methods considered earlier is that it allows us to control the number of weights without altering the overall network structure. We define the *connectivity* of a sparse model to be the ratio between the number of its parameters and the number of parameters in a dense model of the same width.

In order to implement the sparsity method, we need to choose how to distribute the sparsity both across network layers and within each layer. To prevent smaller layers from being cut out entirely, we choose the number of weights to be removed in each layer to be proportional to the number of weights in that layer (except for BatchNorm layers, which are kept intact). Figure 4 demonstrates the principle of our algorithm for sparsity distribution. Within each layer, we distribute the sparsity uniformly across all dimensions of the weight tensor. Overall, this method of distributing the weights led to the best performance among the different methods we tried in our experiments (see Appendix C for more details).

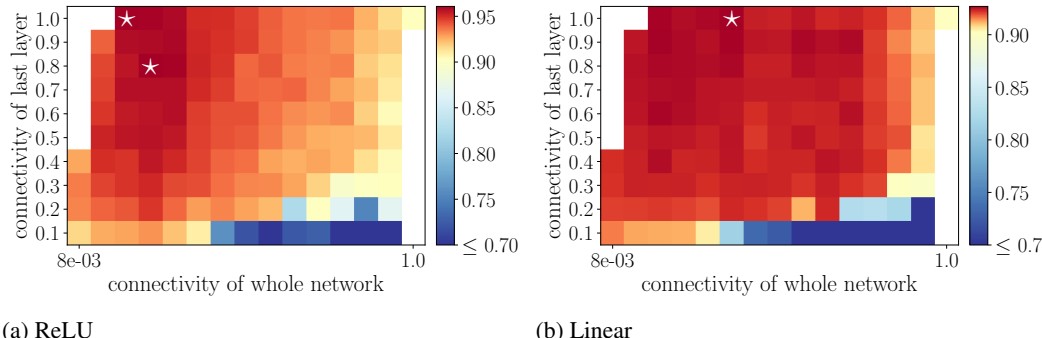

(a) ReLU                                                    (b) Linear

Figure 5: Test accuracy (color-coded) on MNIST obtained by MLP models with 1 hidden layer and ReLU or linear activations, as a function of overall connectivity (horizontal axis) and of connectivity in the last layer (vertical axis). White stars indicate points with the highest test accuracy.

**MLP.** We study a fully-connected network with one hidden layer trained on MNIST. We use this simple setup to compare different sparsity distribution patterns across layers for a given network connectivity, and to compare the effects of increased width with and without ReLU non-linearities.

The dense MLP at width $n$ has two weight matrices: The first layer matrix has size $784 \times n$ and the last layer has size $n \times 10$. In the experiments, we set the total number of weights to 3970 and consider widths ranging from 5 to 640, corresponding to network connectivities between 1 and $5/640 \approx 0.008$. At each width, we vary the connectivity of the last layer (the smaller of the two) between 1.0 and 0.1.

The results are shown in Figure 5. We find that sparse, wide models can outperform the dense, baseline models for both ReLU and linear activations. The ReLU model attains its maximum performance at around $3 - 6\%$ connectivity, which corresponds to a widening factor of 16 or 32. At the optimal point the connectivity of the last layer is high, above $80\%$. It is therefore advantageous in this case to remove more weights from the first layer than from the last layer. This makes intuitive sense: Removing weights from a layer that starts out with fewer weights can be expected to make optimization more difficult. This result motivates our choice to remove weights proportionally to layer size when sparsifying other models. Finally, the fact that larger width leads to improvement even in the deep linear model implies that the improvement cannot be attributed only to the increased model capacity that a wider ReLU network enjoys.

**ResNet-18.** We train families of ResNet-18 models on ImageNet, CIFAR-10, CIFAR-100 and SVHN, covering a range of widths and model sizes. A detailed example of the sparsity distribution over all layers is shown in Appendix D.

Table 1 shows the results obtained on ImageNet. As expected, the performance improves as the width and the number of weights increase (row 1). However, up to a certain width, a comparable improvement is achieved when only the width grows and the number of weights remains fixed (row 2). We point out that test accuracy declines around the same width that training accuracy declines (see additional plots presented in Appendix E). Therefore, in this case the determining factor for model performance is width rather than number of parameters. Figure 6 shows the results of training model families on several additional datasets. We again find that performance improves with width at a fixed number of parameters, up to a certain widening factor. The effect is most pronounced for more difficult tasks and for smaller models that do not reach 100% training accuracy, yet it is still present for models that do fit the training set (see also Appendix E).

| width | 64 | 90 | 128 | 181 | 256 |
|---|---|---|---|---|---|
| **dense** | 68.03 (11.7) | 69.11 (22.8) | 70.22 (45.7) | 70.91 (90.7) | 71.89 (180.6) |
| **sparse** | – | 69.56 (11.7) | 70.02 (11.7) | 70.66 (11.7) | 70.53 (11.7) |

Table 1: ResNet-18 on ImageNet: Top-1 test accuracy (in %) and in parentheses the number of weights in millions. All sparse models have the same number of weights as the smallest dense model, yet the improvement obtained with increasing width is on par with the dense models, up to a certain width.

Figure 7 compares the performance improvement of a sparse, wide model against that of a dense model with the same width. In particular, it shows the fraction of the improvement that can be attributed to width alone: That is, the ratio between the sparse/baseline accuracy gap and the dense/baseline accuracy gap. We see that as long as the model can achieve high training accuracy, most of the improvement in performance can be attributed to the width.

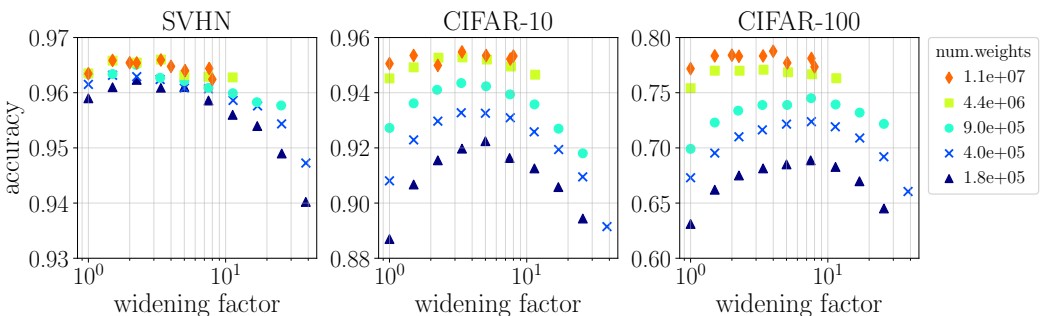

Figure 6: Best test accuracy obtained by ResNet-18 models of different width and size (i.e., total number of weights; approximate value shown in the legend). The leftmost data point of each color corresponds to the dense baseline model, and all subsequent data points correspond to its wider and sparser variants. The decline of performance at larger widening is because sparsity at these levels harms the optimization procedure, making the training accuracy deteriorate. See Appendix E for the same data plotted as a function of network connectivity, the corresponding training accuracy, and for the version with error bars.

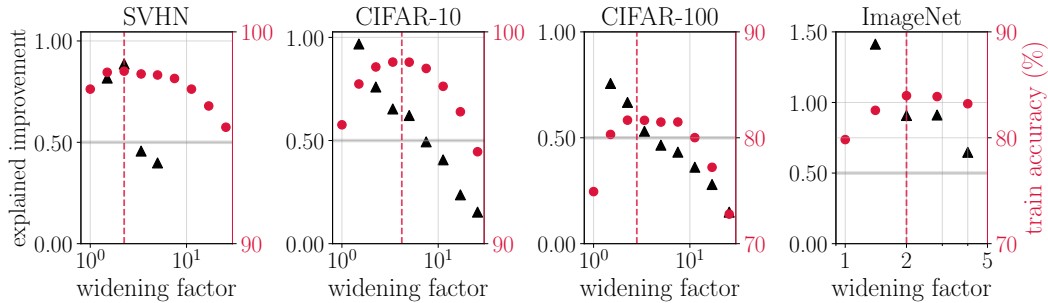

Figure 7: The fraction of test accuracy improvement (black triangles) obtained by widening the model without increasing the number of weights, compared to increasing the number of weights along with the width. The dashed vertical line indicates widening factor for maximal training accuracy (red circles). Note that for ImageNet, at width 90 (widening factor 1.4) the sparse model attained a higher test accuracy than the dense model, as reported in Table 1.

## 3 THEORETICAL ANALYSIS IN A SIMPLIFIED SETTING

We showed empirically that wide, sparse networks with random connectivity patterns can outperform dense, narrower networks when the number of parameters is fixed. The performance as a function of the width has a maximum when the network is sparse. In this section we investigate a potential theoretical explanation for this effect.

It is well known that wider (dense) networks can achieve consistently better performance. In the infinite-width limit, the training dynamics of neural networks is equivalent under certain conditions to kernel-based learning. In particular, consider a network function $f_\theta(x)$ with model parameters $\theta$. The Neural Tangent Kernel is defined by $\Theta(x_1, x_2) := \nabla_\theta f(x_1)^T \nabla_\theta f(x_2)$. Jacot et al. (2018) showed that, in the infinite width limit (and with appropriate parameterization defined below), training the network $f_\theta$ using gradient flow is equivalent to training a linear model with the kernel $\Theta$. Lee et al. (2019) showed empirically that the NTK lineraization result can be a good approximation for the training dynamics of networks with large but finite width.

Based on these results, we propose a possible explanation for the main effect observed in this paper, which is valid for networks with sufficiently large but finite width. We conjecture that the kernel of a

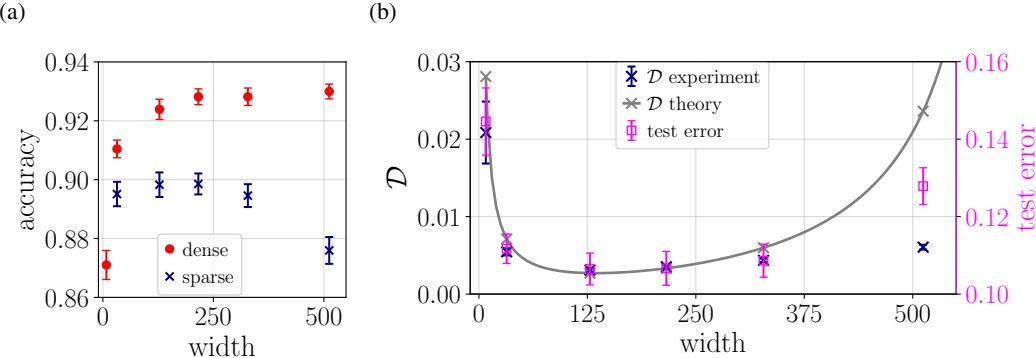

Figure 8: MLP with 1 hidden layer and no biases trained on a subset of MNIST. (a) Test accuracy achieved by dense (filled circles) and sparse (crosses) models of different width. (b) Mean squared distance $\mathcal{D}$ of the sparse model's kernel from the infinite-width model's kernel computed at initialization (both experimental (blue) and theoretical (grey) result), and the test error attained by trained models (pink). The empirical distance $\mathcal{D}$ is obtained by averaging the squared distance $(\Theta_{\mathrm{GP}}(x,y) - \Theta_{\mathrm{GP}}^{\infty}(x,y))^2$ over $10^4$ pairs of test samples and over 10 random initializations. See Appendix A for additional details.

finite-width network at initialization is indicative of its performance, and that optimal performance is achieved when its distance to the infinite-width kernel is minimized. We further hypothesize that this distance can be reduced by increasing the network width at a fixed number of parameters. In the following, we formalize this conjecture and derive expressions for the kernel of a sparse finite-width network with one hidden layer. We calculate the kernel distance theoretically, and show that the distance predicted using this result is in good agreement with experiments.

Consider a 2-layer ReLU network function $f : \mathbb{R}^d \to \mathbb{R}$ given by $f(x) = (nd)^{-1/2} v^T [ux]_+$, $[z]_+ := zH(z)$. Here, $x \in \mathbb{R}^d$ is the input, the network parameters are $u \in \mathbb{R}^{n \times d}$ and $v \in \mathbb{R}^n$ (for simplicity, we omit the biases), and $H(\cdot)$ is the Heaviside step function. We note that in this section we use NTK parameterization (Jacot et al., 2018) for simplicity, whereas in previous sections we used LeCun parameterization. Each network parameter is sampled independently from $\mathcal{N}(0, \sigma^2)$ with probability $p$, and is set to zero with probability $1 - p$. We set $\sigma^2 = p^{-1}$. The probability density function for each parameter $\theta$ is therefore given by $\Pr(\theta) = p(2\pi\sigma^2)^{-1/2} \exp\left(-\theta^2/2\sigma^2\right) + (1-p)\delta(\theta)$, where $\delta(\cdot)$ is the Dirac delta function. In the following, we consider the Gaussian Process (GP) kernel of the network, defined as $\Theta_{\mathrm{GP}}(x,y) := \nabla_v f(x)^T \nabla_v f(y)$.

**Theorem 1.** *Consider the GP kernel $\Theta_{\mathrm{GP}}$ of a 2-layer network with ReLU activations, where the weights are sampled from $\Pr(\theta)$. The mean distance squared between $\Theta_{\mathrm{GP}}$ to the (dense) infinite-width kernel $\Theta_{\mathrm{GP}}^{\infty}$ is*

$$\mathbb{E}_\theta\left[(\Theta_{\mathrm{GP}}(x,y) - \Theta_{\mathrm{GP}}^{\infty}(x,y))^2\right] = \frac{1}{d^2}\left[\tilde{K}_{1,p}(x,y) - K_1(x,y)\right]^2 + \frac{1}{d^2 n}\left[\tilde{K}_{2,p}(x,y) - \tilde{K}_{1,p}(x,y)^2\right].$$
(1)

*Here,*

$$\tilde{K}_{l,p}(x,y) = \sigma^{2l} \cdot \sum_{s_1,\dots,s_d \in \{0,1\}} p^{\sum_i^d s_i}(1-p)^{d-\sum_i^d s_i} K_l(x_s, y_s),$$
(2)

$$K_l(x,y) = \frac{1}{(2\pi)^{d/2}} \int_{-\infty}^{\infty} d^d w \, e^{-\|w\|_2^2/2} (w \cdot x)^l (w \cdot y)^l H(w \cdot x) H(w \cdot y).$$
(3)

$s = (s_1, \dots, s_d)$ *is a vector with elements in $\{0,1\}$, and $x_s := (s_1 x_1, \dots, s_d x_d)$ is the $x$ vector with some elements zeroed out.*

The proof is provided in Appendix F. Also in the Appendix, we derive the following closed-form approximation to the kernel distance (1) under the assumptions that the input vectors $x, y$ are independent random vectors and $pd \gg 1$:

$$\mathbb{E}_\theta\left[(\Theta_{\mathrm{GP}}(x,y) - \Theta_{\mathrm{GP}}^{\infty}(x,y))^2\right] \approx \frac{1}{4d}\left[\frac{1}{4}\left(\frac{1}{\sqrt{p}} - 1\right)^2 + \frac{d}{n}\left(1 - \frac{1}{\pi^2}\right)\right].$$
(4)

In order to keep the number of parameters fixed as we change the width, we set $np = $ const. Under this constraint, and assuming $n \gg 1$, the distance (4) is minimized when $p_* \approx \sqrt{np/4d}$.

In Figure 8b we compare this approximation with the GP kernel computed empirically at initialization. We find good agreement with the theoretical prediction (4) when $dp \gg 1$. Furthermore, we see that the minimal kernel distance at initialization and the optimal performance of the trained network are obtained at a similar width, providing evidence in support of our hypothesis.

## 4 DISCUSSION

In this work we studied the question: Do wider networks perform better because they have more parameters, or because of the larger width itself? We considered several ways of increasing the width while keeping the number of parameters fixed, either by introducing bottlenecks into the network, or by sparsifying the weight tensors using a static, random mask. Among the methods we tested, the one that led to the best results was removing weights at random in proportion to the layer size, using a static mask generated at initialization. In our image classification experiments, increasing the width using this sparsity method (while keeping the total number of parameters constant) led to significant improvements in model performance. The effect was strongest when starting with a narrow basline model. Additionally, when comparing the wide, sparse models against dense models of the same width, we found that the width itself accounts for most of the performance gains; this holds true up to the width above which the training accuracy of the sparse models begins to deteriorate, presumably due to low connectivity between neurons.

Focusing on the sparsity method, we initiated a theoretical study of the effect, hypothesizing that the improvemenet in performance is correlated with having a Gaussian Process kernel that is closer to the infinite-width kernel. We computed the GP kernel of a sparse, 2-layer ReLU network, and derived a simple approximate formula for the distance between this kernel and the infinite-width dense kernel. In our experiment, we found surprisingly strong correlation between the model performance and the distance to the infinite-width kernel.

While our work is fundamental in nature, and sparsity is not the subject of this paper, the method we propose may lead to practical benefits in the future. Using current hardware and available deep learning libraries, we cannot reap the benefits of a sparse weight tensor in terms of reduced computational budget. However, in our experiments we find that the optimal sparsity can be around 1-10% for convolutional models (corresponding to a widening factor of between 3-10). Therefore, using an implementation that natively supports sparse operations, our method may be used to build faster, more memory-efficient networks.

## ACKNOWLEDGEMENTS

The authors would like to thank Ethan Dyer, Utku Evci, Etai Littwin, Joshua Susskind, and Shuangfei Zhai for useful discussions.

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

## A  EXPERIMENTAL DETAILS

In this section we provide experimantal details and additional information about the figures in the main text.

**Figures 1 and 6:**   In all experiments, we use a standard PyTorch implementation of the ResNet-18 model. We set the number of weights in the model by changing the number of output channels in the first convolutional layer (referred to as *the* model width), while leaving the width ratios between the first convolutional layer and the subsequent four blocks of the ResNet-18 at their default values $1:2:4:8$. We do not include the weights in the BatchNorm layers into the total weight count, and we do not sparsify these layers.

All models are trained using SGD with momentum=0.9, Cross-Entropy loss, and initial learning rate 0.1. The learning rate value and schedule were tuned for the smallest baseline model. We do not apply early stopping, and we report the best achieved test accuracy. For ImageNet, we use weight decay 1e-4, cosine learning rate schedule, and train for 150 epochs. Because of the computational cost of ImageNet experiments, we did not repeat each run multiple times. We have checked for two datapoints that the variance in training and test accuracy for different random seeds is smaller than 0.1%. For other datasets, we use weight decay 5e-4, train for 300 epochs, and the initial learning rate 0.1 is decayed at epochs 50, 120 and 200 with gamma=0.1. The reported results are averages over 3 runs with different random seeds.

In Figure 1, the baseline model has width 64 (1e7 weights) for ImageNet, and 18 (9e5 weights) for the other datasets.

In Figure 6, we consider baseline models with base widths [8, 12, 18, 40, 64], corresponding to a total of [1.8e5, 4.0e5, 9.0e5, 4.4e6, 1.1e7] weights respectively.

**Figure 5:**   All networks are MLPs with one hidden layer, a total of 3970 weights (base width is 5), and either (a) ReLU or (b) Linear activation function. The networks are parameterized according to the standard Pytorch implementation (weights and biases are randomly initialized from the uniform distribution). We train these models on MNIST for a fixed number of 300 epochs (ensuring convergence), with SGD optimizer, no momentum, Cross-Entropy loss, with a constant learning rate 0.1 and mini-batch size 100.

For ReLU, highest test accuracy (marked by white stars in the plot) is 96.3%, and is achieved by models with connectivity 0.06 (width 80) or 0.03 (width 160). For Linear, it is 92.7%, achieved at connectivity 0.13 (width 40). The color scheme is centered at the test accuracy value attained by the baseline model (approximately 90% in both cases), and its upper limit is set to the respective highest achieved value. Empty (white) cells correspond to invalid combinations of connectivity values. Note that the range on the horizontal axis is not equally spaced.

**Figure 8:**   The MLP has one hidden layer, no biases, ReLU activation function, and NTK-style parameterization. It is trained on a subset of 2048 samples from the MNIST training set and tested on the full MNIST test set. The input is normalized with pixel mean and standard deviation as `(image − mean)/stdev`. We train for 300 epochs with vanilla SGD using Cross-Entropy loss and batch size 256. The learning rate was tuned separately for each width. The reported numbers are averages over 10 random seeds.

The number of weights in dense models is $(784 + 10) \cdot width$, while all sparse models have the same number of weights as the smallest dense model (width 8): 6,352. The empirical approximation of the infinite-width kernel is computed on a dense MLP with width $10^4$ at initialization.

### A.1  IMAGENET DATA PREPROCESSING

In order to decrease the size of the dataset and be able to download it to cloud instances with limited storage, we resized all images in the dataset by keeping their proportions fixed and setting their smallest dimension to 256. This procedure reduces the accuracy of ResNet models by aboud 1-2%.

**Transformations for ImageNet:**   We follow the standard transformations used is training Imagenet. Following is the list of PyTorch data transformations applied on each image.

- `RandomResizedCrop(size=224, scale=(0.2, 1.0))` on the training set.
- `Resize(256, transforms.CenterCrop(224))` on the test set.

## B  SPARSITY DISTRIBUTION CODE

The following code implements our algorithm for distributing sparsity over model layers. Figure 4 illustrates the procedure.

```python
def get_ntf(num_to_freeze_tot, num_W, tensor_dims, lnames_sorted):
    """ Distribute the total number of weights to freeze over model layers.

    Parameters
      num_to_freeze_tot (int) - total number of weights to freeze.
      num_W (dict) - layer names (keys) and number of weights in layer
↪   (vals).
      tensor_dims (dict) - layer names (keys) and the dimensions of layer
↪   tensor (vals).
      lnames_sorted (list of str) - layer names, sorted by magnitude in
↪   descending order.

    Returns
      num_to_freeze (list of int) - number of weights to freeze per layer,
↪   order corresponding to lnames_sorted.
    """

    num_layers = len(lnames_sorted)
    num_to_freeze = np.zeros(num_layers, dtype=int) # init

    # list of num. weights in layer, in sorted order (largest first)
    num_W_sorted_list = [num_W[lname] for lname in lnames_sorted]

    # compute  num. weights differences between layers
    num_W_diffs = np.diff(num_W_sorted_list)
    num_W_diffs = [abs(d) for d in num_W_diffs]

    # auxiliary vector for the following dot product to compute the bins
    aux_vect = np.arange( 1,len(num_W_diffs)+1 )

    # the bins of the staggered sparsification: array of max. num. of
↪   weights that can be frozen within the given layer before the
↪   next-smaller layer gets involved into sparsification
    ntf_lims = [np.dot(aux_vect[:k], num_W_diffs[:k]) for k in
↪   range(1,num_layers)]

    # find in which bin num_to_freeze_tot falls - this gives the number of
↪   layers to sparsify
    lim_val, lim_ind = find_ge(ntf_lims, num_to_freeze_tot)
    num_layers_to_sparsify = lim_ind+1

    # base fill: chunks of num. weights that are frozen in each involved
↪   layer until all involved layers have equal num. weights remaining
    base_fill = [sum(num_W_diffs[lind:lim_ind]) for lind in range(lim_ind)]
    base_fill.append(0)

    # the rest is distributed evenly over all layers involved
    rest_tot = num_to_freeze_tot-sum(base_fill)
    rest = int(np.floor(rest_tot/num_layers_to_sparsify))
    num_to_freeze[:num_layers_to_sparsify] = np.array(base_fill)+rest

    # first layer gets the few additional frozen weights when rest_tot is
↪   not evenly divisible by num_layers_to_sparsify
    rest_mismatch = rest_tot - rest*num_layers_to_sparsify
    num_to_freeze[0]+= rest_mismatch
```

```
    assert sum(num_to_freeze)==num_to_freeze_tot

    return num_to_freeze
```

## C  SPARSITY DISTRIBUTION IN CONVOLUTIONAL LAYERS

The weight tensor of a convolutional layer has 4 dimensions: input, output, kernel width, kernel height. As discussed in section 2.3, when reducing network connectivity, we remove weights randomly across all dimensions of a weight tensor. A reasonable alternative for convolutional layers is to remove weights in the input and output dimensions only, leaving the kernels themselves unchanged. We test this approach on ResNet-18 and find that it leads to very similar results in general. Specifically for smaller models, removing weights along all tensor dimensions results in better performance. For the smallest networks (base widths 8 and 12) and small connectivity, we observed a gap up to 3% in test accuracy on both CIFAR datasets. Figure 9 shows results for on CIFAR-100, where the effect is more pronounced, for models with base widths 8, 12 and 18 (1.8e5, 4.0e5 and 9.0e5 weights, respectively). Each experiment was repeated for 10 random seeds.

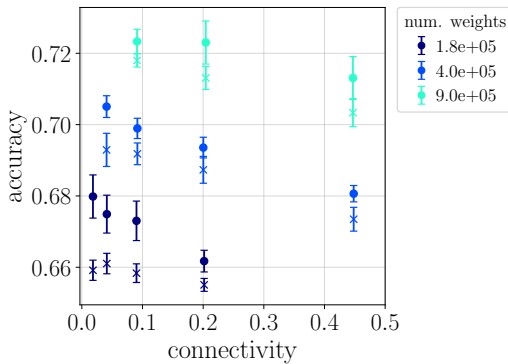

Figure 9: Test accuracy achieved by ResNet-18 models on CIFAR-100, comparing performance of sparse wide models of different size (number of weights indicated by color and printed in the legend) given sparsity distribution in the convolutional layers along all layer dimensions (filled circle) versus along input/output dimensions only (cross).

## D  SPARSITY DISTRIBUTION IN RESNET-18

On a coarse level, the ResNet-18 architecture is as follows: one convolutional layer, followed by four modules, followed by one fully-connected layer; each module comprises two blocks, and each block contains two convolutional layers. The number of output channels in the first convolutional layer is the same for the first module, and the ratio of output channel numbers in the subsequent modules is $1 : 2 : 4 : 8$ – we do not change this ratio in our experiments. When building a family of ResNet-18 models, we vary the number of output channels of the first convolutional layer, and refer to this as *the width* of the model, while the widths of all subsequent layers are set according to the mentioned ratio.

When reducing the connectivity of a ResNet-18 model, we remove weights from each layer according to layer size. More precisely, we first remove weights from the layer with the largest number of weights until it reaches the size of the next-smaller layer. We then proceed with removing weights from these two layers equally, and continue this procedure until the targeted total number of weights in the network is achieved.

Figures 10 shows layer-wise sparsity distribution in ResNet-18 with 1.8e5 weights and various widths as an example.

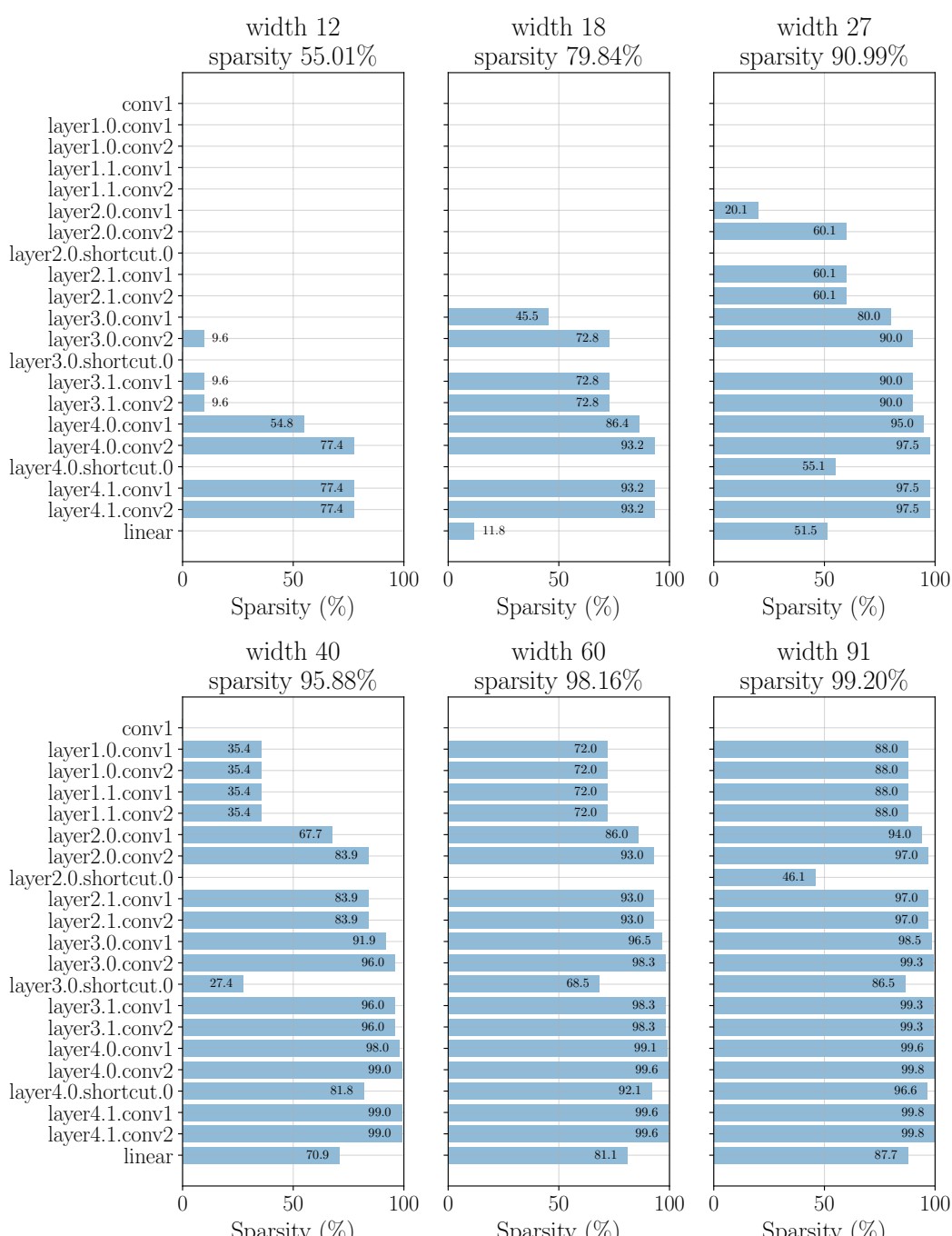

Figure 10: Layer-wise sparsity distribution in ResNet-18 of various widths with 1.8e5 weights, for the CIFAR-100 dataset.

# E   ADDITIONAL FIGURES FOR RESNET-18 EXPERIMENTS

In this section we show additional plots for ResNet-18 experiments – see Figures 11, 12, 13, and 14.

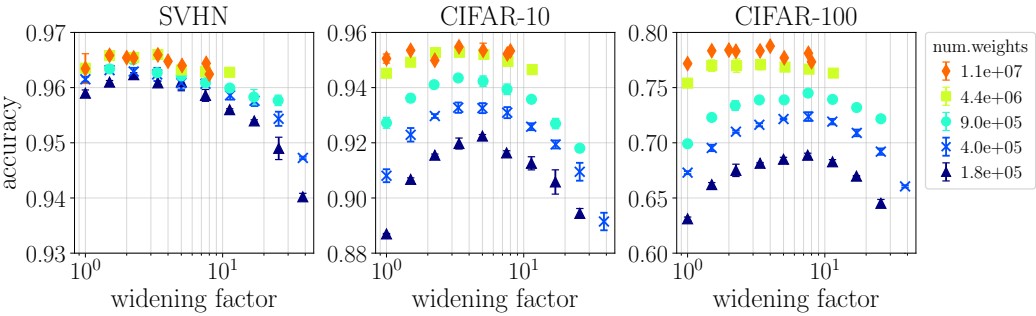

Figure 11: Same data as in Figure 6, but with error bars.

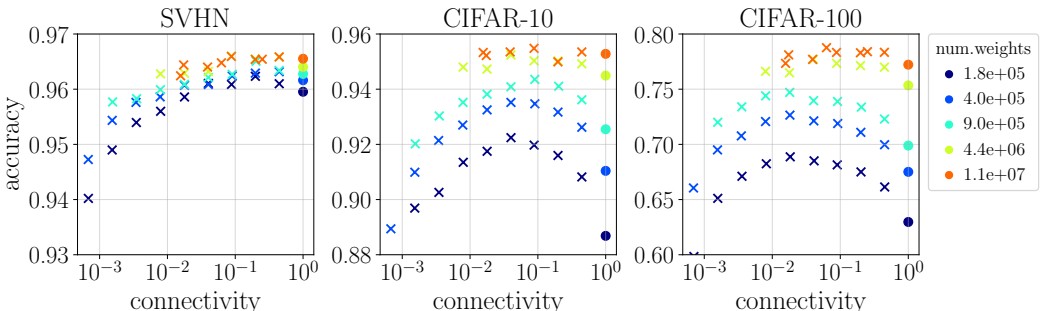

Figure 12: Same data as in Figure 6, but plotted as a function of network connectivity instead of width. Best test accuracy obtained by ResNet-18 models of different width (number of output channels of the first convolutional layer) and size (total number of weights, values indicated by marker color and shown in the legend). The rightmost data point of each color (filled circle) corresponds to the dense baseline model (connectivity 1), all other data points (crosses) correspond to its wider and sparser variants. For smaller models (up to 9.0e+5 weights), the performance peaks at similar connectivity values.

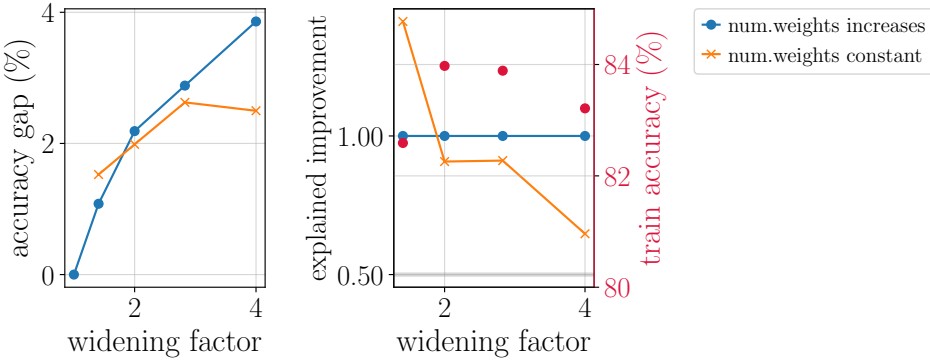

Figure 13: The improvement in test accuracy due to widening, with constant or increasing number of weights. Data from the same experiments as in Table 1.

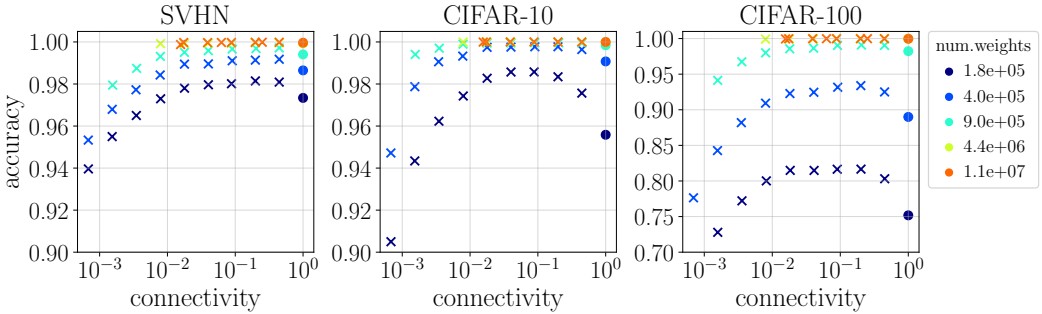

Figure 14: Training accuracy from the same experiments as in Figures 6 and 12. Larger models attain 100% training accuracy. For smaller models, the training accuracy shows a similar behavior as the test accuracy: It increases up to a certain connectivity and then deteriorates when the connectivity decreases further.

## F  THEORETICAL DETAILS

In this section we provide additional details on the theoretical analysis of Section 3.

*Proof (Theorem 1).* We begin by defining the integral

$$
\tilde{K}_{l,p}(x,y) := \int d^d w \prod_i^d \Pr(w_i)(w \cdot x)^l (w \cdot y)^l H(w \cdot x) H(w \cdot y). \tag{5}
$$

In particular, for $p = 1$ we have the relation $\tilde{K}_{l,p}(x,y) = K_l(x,y)$.

A straightforward calculation gives the following.

$$
\tilde{K}_{l,p}(x,y) = \sum_{s_1,\dots,s_d \in \{0,1\}} p^{\Sigma_s}(1-p)^{d-\Sigma_s}
$$

$$
\times (2\pi\sigma^2)^{-\Sigma_s/2} \int dw_s\, e^{-|w_s|^2/2\sigma^2}(w_s \cdot x)^l (w_s \cdot y)^l H(w_s \cdot x) H(w_s \cdot y) \tag{6}
$$

$$
= \sigma^{2l} \cdot \sum_{s_1,\dots,s_d \in \{0,1\}} p^{\Sigma_s}(1-p)^{d-\Sigma_s} K_l(x_s, y_s). \tag{7}
$$

Here $s = (s_1, \dots, s_d)$ is a 0,1 vector of length $d$, $\Sigma_s = \sum_i^d s_i$, and $w_s = (s_1 w_1, \dots, w_d s_d)$ is the $w$ vector with some elements zeroed out ($x_s$ and $y_s$ are defined similarly). The integration is over all the non-zero $w$s, namely $dw_s = \prod_{i|s_i=1} dw_i$.

Consider now the sparse kernel $\Theta_{\mathrm{GP}}$. It is easy to check that

$$
\mathbb{E}_\theta[\Theta_{\mathrm{GP}}(x,y)] = \frac{1}{d}\tilde{K}_{1,p}(x,y), \tag{8}
$$

$$
\mathrm{Var}[\Theta_{\mathrm{GP}}(x,y)] = \frac{1}{d^2 n}\left[\tilde{K}_{2,p}(x,y) - \tilde{K}_{1,p}(x,y)^2\right]. \tag{9}
$$

Let $\Theta_{\mathrm{GP}}^n$ be the dense kernel (with $p = 1$) at width $n$, and let $\Theta_{\mathrm{GP}}^\infty = \lim_{n\to\infty} \Theta_{\mathrm{GP}}^n$ be the dense infinite-width kernel. From (8) and (9) we see that $\mathbb{E}_\theta[\Theta_{\mathrm{GP}}^\infty(x,y)] = \frac{1}{d}K_1(x,y)$, and $\mathrm{Var}[\Theta_{\mathrm{GP}}^\infty(x,y)] = 0$. Using these results, the mean square distance between the sparse and infinite-width kernels is now given by

$$
\mathbb{E}_\theta\big[(\Theta_{\mathrm{GP}}(x,y) - \Theta_{\mathrm{GP}}^\infty(x,y))^2\big] = [\mathbb{E}_\theta[\Theta_{\mathrm{GP}}(x,y)] - \Theta_{\mathrm{GP}}^\infty(x,y)]^2 + \mathrm{Var}[\Theta_{\mathrm{GP}}(x,y)]
$$

$$
= \frac{1}{d^2}\left[\tilde{K}_{1,p}(x,y) - K_1(x,y)\right]^2 + \frac{1}{d^2 n}\left[\tilde{K}_{2,p}(x,y) - \tilde{K}_{1,p}(x,y)^2\right]. \tag{10}
$$

$\square$

### F.1 APPROXIMATING THE KERNEL DISTANCE

Next, we derive the approximate form (4) by using plausible arguments. In this calculation we assume that $dp \gg 1$, and that $x, y \in \mathbb{R}^d$ are independent random vectors with elements sampled from $\mathcal{N}(0, 1)$. The derivation is not rigorous, but we compare the results against an empirical calculation in Figure 8 and find good agreement when $dp \gg 1$.

For given $p$ we expect the dominant contribution to $\tilde{K}_l$ in the sum (7) to come from terms where $\Sigma_s = \sum_i s_i \approx pd$, and so we consider a mask $s$ with this property. We can then approximate $x_s \cdot y_s \approx dp$ and $\|x_s\|_2 \approx \sqrt{dp}(1 - 1/4dp)$ (and similarly for $\|y_s\|_2$).[4] We denote by $\theta$ the angle between $x, y$, and by $\theta_s$ the angle between $x_s, y_s$. Then, from the above we expect

$$\cos \theta_s \approx \frac{\xi}{\sqrt{dp}}, \quad \sin \theta_s \approx \sqrt{1 - \frac{1}{dp}} \approx 1 - \frac{1}{2dp}, \quad \theta_s \approx \frac{\pi}{2} - \frac{\xi}{\sqrt{dp}}, \tag{11}$$

where $\xi = \pm 1$ is a random sign.

Next, we consider the integrals $K_l(x, y)$. We will rely on the following result from Cho and Saul (2009).[5] For $x, y, w \in \mathbb{R}^d$,

$$K_l(x, y) = \frac{1}{(2\pi)^{d/2}} \int d^d w \, e^{-\|w\|_2^2/2} (w \cdot x)^l (w \cdot y)^l H(w \cdot x) H(w \cdot y) \tag{12}$$

$$= \frac{1}{2\pi} |x|^l |y|^l J_l(\theta(x, y)), \tag{13}$$

$$\theta(x, y) = \arccos\left(\frac{x \cdot y}{|x||y|}\right), \tag{14}$$

$$J_1(\theta) = \sin \theta + (\pi - \theta) \cos \theta, \tag{15}$$

$$J_2(\theta) = 3 \sin \theta \cos \theta + (\pi - \theta)(1 + 2 \cos^2 \theta). \tag{16}$$

Using the random vector approximations, we find

$$K_1(x_s, y_s) = \frac{1}{2\pi} |x_s||y_s| J_1(\theta_s) \approx \frac{dp}{2\pi} \left[1 + \frac{\pi \xi}{2\sqrt{dp}}\right], \tag{17}$$

$$K_2(x_s, y_s) = \frac{1}{2\pi} |x_s|^2 |y_s|^2 J_2(\theta_s) \approx \frac{(dp)^2}{2\pi} \left[\frac{\pi}{2} + \frac{4\xi}{\sqrt{dp}} + \frac{\pi}{dp} + \frac{\xi}{2(dp)^{3/2}}\right]. \tag{18}$$

For the sparse functions $\tilde{K}_l(x, y)$, we assume as before that the contribution from the sum over masks is concentrated where the mask $\Sigma_i s_i \approx dp$. For a mask $s$ obeying this condition we then have $\tilde{K}_{l,p}(x, y) \approx \sigma^{2l} K_l(x_s, y_s)$. In particular,

$$\tilde{K}_{1,p}(x, y) \approx \frac{d}{2\pi} \left[1 + \frac{\pi \xi}{2\sqrt{dp}} + \frac{1}{2dp}\right], \tag{19}$$

$$\tilde{K}_{2,p}(x, y) \approx \frac{d^2}{2\pi} \left[\frac{\pi}{2} + \frac{4\xi}{\sqrt{dp}} + \frac{\pi}{dp} + \frac{\xi}{2(dp)^{3/2}}\right]. \tag{20}$$

We can also consider the diagonal elements, setting $x = y$ and $\theta_s = \theta = 0$. We have $\tilde{K}_{l,p}(x, x) \approx \sigma^{2l} K_l(x_s, x_s) \approx \frac{\sigma^{2l} (dp)^l}{2\pi} \left[1 + \frac{l(l-1)}{dp}\right] J_l(0)$. In particular,

$$\tilde{K}_{1,p}(x, x) \approx \frac{d}{2}, \tag{21}$$

$$\tilde{K}_{2,p}(x, x) \approx \frac{3d^2}{2} \left(1 + \frac{2}{dp}\right). \tag{22}$$

---

[4]Here we used the fact that these are random vectors of effective dimension $\approx dp$, and that for such a vector we have $\mathbb{E}_\theta[\|x_s\|_2^l] = 2^{l/2} \frac{\Gamma((d+l)/2)}{\Gamma(d/2)}$.

[5]The definition here has a factor of 2 difference compared with Cho and Saul (2009).

We are now ready to derive the approximate kernel distance (4) from the exact relation (1). Using the approximations (19), (20), (21), (22), we end up with the following.

$$\mathbb{E}_\theta\big[(\bar{\Theta}_{n,p}(x,y) - \bar{\Theta}_\infty(x,y))^2\big] \approx \frac{1}{4d}\left[\frac{1}{4}\left(\frac{1}{\sqrt{p}} - 1\right)^2 + \frac{d}{n}\left(1 - \frac{1}{\pi^2}\right)\right], \tag{23}$$

$$\mathbb{E}_\theta\big[(\bar{\Theta}_{n,p}(x,x) - \bar{\Theta}_\infty(x,x))^2\big] \approx \frac{1}{n}\left(\frac{5}{4} + \frac{3}{dp}\right). \tag{24}$$

We would like to find the minimum of the distance (23) when keeping the mean number of parameters $np$ fixed. Roughly, we would like to minimize $\frac{1}{4}(p^{-1/2} - 1)^2 + \frac{dp}{np}$ with $np$ constant. Assuming that the minimum is at $\sqrt{p} \ll 1$, we find that the minimum $p_*$ is at

$$p_* \approx \sqrt{\frac{np}{4d}}. \tag{25}$$

