# OpenReview forum: "Are wider nets better given the same number of parameters?"
_ICLR.cc/2021/Conference — ICLR 2021 Poster_

### Official Review · AnonReviewer2 · 2020-10-25
**Interesting but a bit vaguely-defined problem**

**Rating:** 4
**Confidence:** 4

**Review:**

Summary:

This paper analyzed the influence of neural network width on the network performances while fixing the total number of parameters. Specifically, the authors introduced several ways to change the model width without increasing the number of parameters, and showed experimentally that for widened networks with a random static mask on weights to keep the number of parameters, increasing the width can improve the performances of the models until the network become very sparse and hard to train. They author theoretically showed that for a not so sparse one-hidden-layer neural network, increasing width decreases the distance to the Gaussian Process kernel corresponding to the infinith-width limit, which partially supports their experimental findings.

Pros:

1. Understanding the reason behind the generalization performance of neural networks is a very important problem, and the idea of decoupling the effects of network width and the number of parameters on the generalization performance is interesting.

2. This paper is clearly written, well organized, and generally easy to understand. The figures in this paper are clear, and each figure was explained in detail. The methods/experimental settings are explained in detail, and the reasons why the authors use these models/methods are also discussed in the paper. The relationship to prior works has been clearly discussed.

3. The experimental methodologies and theoretical computations appear to be correct.

Cons:

1. The justification for this specific problem analyzed in this paper, i.e., fixing the number of parameters and changing the width, is a bit unclear. The number of parameters can be interpreted as a measure for the expressive power of the model, but this is not a very direct relationship because the expressive power is also related to other factors, e.g., the depth, the width, the choice and ordering of the layers. In the methods proposed in this paper, the authors were either changing the explicit number of parameters (e.g., linear bottlenetwork) or changing the expressive power of the network, which included other (perhaps unwanted) factors in this framework and could potentially influence the results. Details will be discussed in the next point.

2. The definition of "number of parameters" is a bit unclear in this paper, making the problem that the authors want to analyze a bit vague. The authors didn't formally define it, but it is actually one of the core concept in this paper. The authors used three similar terms, i.e., "number of parameters"/"number of tunable parameters", "effective number of parameters", and "expressive power", in this paper. However, these three terms have slightly different meanings, and none of them is defined in this paper. From my understanding, "expressive power" is the set of all functions that can be reprensented by a model, and "number of parameters"/"number of tunable parameters" is a simple parameter count and can be used as a complexity measure of the expressive power, but "effective number of parameters" is kind of confusing. For instance, putting a low-rank constraint in Linear Bottleneck or using a sparse model actually changes the expressive power of the neural network but (claimed by this paper) does not change the "effective number of parameters". Due to the lack of formal definition of these three terms, I think the problem analyzed by this paper is not clear enough, making the points made in this paper unclear.

3. The authors are explaining some of the experimental results in a very intuitive way without enough supporting evidence. For example, the authors said that Linear Bottleneck suffers from some "implicit bias", deleting weights from a sparse layer make optimization "more difficult", and small connectivity can hurt the "trainability" of the model. These terms are not formally defined or explained in detail, making the points made by the authors a little vague.

4. The connection between the theory and experiments in this paper is a bit weak. People have observed that the performances of commonly-used neural networks always outperform their infinith-width limit, i.e., the corresponding infinith-width NTK. Therefore, in practical neural network models, simply decreasing the distance to the infinith-width limit may hurt the performance. Thus, I suspect that the explanation provided by the authors only works for some particular examples like small networks on MNIST.

Recommendation:

Overall, I vote for rejecting this paper. Although the idea of decoupling the influence of width and number of parameters is interesting, the problem analyzed in this paper is vague and a lot of terms used in this paper is not properly defined, making the arguments in this paper unclear.

Supporting arguments for recommendation:

See "Cons".

Questions for the authors:

1. Please address the cons mentioned above.

2. For the experiments, I wonder whether you have done enough fine-tuning for the hyperparameters of each model and used the best test accuracy. Also, have you run each experiment multiple times or only once?

3. I didn't understand why you do not consider the linear/non-linear bottleneck as an effective method. I think the goal is to analyze the effect of width but not to get a significant improvement on the test accuracy.

Minor comments:

1. Typo: Page 5, "MLP", paragraph 3, "This results" -> "This result"

2. Page 5, paragraph 1, which model is "the model with baseline width 8"? Is it the model with the number of parameters 1.8e+5?

---

> ### Author Response · Authors · 2020-11-17
> **Thanks for your detailed review! We have made updates and added clarifications that address all your concerns!**
>
> Thank you for your valuable comments which indeed helped us improve the paper by removing vague terms and arguments.
> Your review suggests that our main idea is important and interesting, our paper is well-written, and our experimental methodologies and theoretical computations are correct. However, your main concern seems to be that some of the terms/discussions in the paper are vague.
> Below, we explain how we have addressed your concern to the best of our ability. However, we want to emphasize that this problem has a complex nature which makes it difficult to obtain a definitive solution. It is impossible to run a completely controlled experiment which allows one to fix #param and change the width while controlling everything else in the architecture and optimization setup. Any empirically chosen modification would lead to changing other factors, many of which are not well-understood, hence making clear arguments difficult. We believe that in our analysis we have done the best possible to isolate and study the effect of interest. We hope that you will consider accepting the paper given the complex nature of this problem, our clear contributions, and our modifications to address your concerns.
>
> Re the problem studied in this paper:
>
> Even though the number of parameters is only an indirect measure of the expressive power of a model, we chose it because the number of parameters has been widely used in papers related to understanding over-parameterization and generalization. In fact, in many theoretical papers, either width or number of parameters appear in the bounds. Therefore, we wanted to disentangle these two factors.
>
> Re the definition of terms:
>
> Thanks for pointing this out. We apologize for the confusion caused by our choice of terms. The usage of “effective number of parameters” and “number of tunable parameters” was not necessary, and we have removed them from the text. We have also defined the term “expressive power” to avoid confusion.
>
> Re explaining results in an intuitive way:
>
> Thank you for pointing this out. We have clarified these explanations in the revised version, avoiding making intuitive claims without supporting evidence.
>
> Re connection between theory and experiment:
>
> We accept the criticism. Our theoretical analysis can provide an explanation for the observed effect for networks that are close to the NTK regime. Networks used in practice are trained outside of this regime, not only because of their finite width but also because of other hyperparameter choices (large learning rates, weight decay, etc.). However, it is often true in deep learning that one can develop theoretical understanding only in a limited setting, and we believe that such understanding can still be valuable in the long term.
>
> Re question 2 -- Hyperparameter tuning and random seeds:
>
> We have fine-tuned the parameters for the baseline models, and kept them fixed for the widened models, which allowed us to avoid 10x more experiments. This makes any claim about improvements over baseline credible, since the chosen hyperparameters are optimized for the baseline.
>
> We did repeat each experiment with multiple random seeds whenever feasible and reported the averages. For MNIST experiments, we used 10 seeds. For CIFAR and SVHN, we averaged over 3 random seeds, except for the largest model size. The standard deviation is small, such that in most cases the error bars are of the size of the plot markers. For reference, we have included a plot with error bars in the Appendix (Section E, Figure 11).
>
> Re question 3 -- Why not consider linear/non-linear bottleneck:
>
> To be clear, in the paper we do consider all three methods, and in the first stage of our study we analyze and compare them. We believe that the answer to the question of decoupling width and number of parameters is this: “it’s complicated”. We do our best to highlight the differences between each of these methods, and the factors that are being changed by choosing any one of them. Finally, we proceed to show that performance can be improved by increasing the width without changing the number of model parameters. Since sparsity is the most promising method based on empirical performance (test accuracy given a certain width), we continue our investigation further on sparsity.

---

### Official Review · AnonReviewer3 · 2020-10-26
**I provide my personal opinions on the analyses about the effect brought by widening nets from both experimental and theoretical perspectives.**

**Rating:** 5
**Confidence:** 3

**Review:**

In this paper, the authors analyze the enhancements brought by widening networks with the number of parameters fixed. From the experimental side, they conduct various experiments to compare the methods of widening the networks and demonstrate different ratios of widening different networks on diverse datasets. From the theoretical side, the authors relate the training dynamics of neural networks to kernel-based learning, in  the infinite-width limit. As a consequence, the authors claim that wider networks indeed improve the performance of algorithms under certain conditions.

My personal comments are listed below.
Pros.
1. The idea of widening the neural networks with fixed number of parameters is novel and the authors conducts abundant experiments to support their assertions.
2. The expressions and grammars are excellent.

Cons.
I. Clarity Part.
1. The authors spend much time and space to introduce linear and non-linear bottleneck methods to widening the neural networks with fixing the number of parameters, and, subsequently, prove the 2 methods are not qualified. From my opinion, since the authors have asserted that they try to avoid big adjustments on structure of the networks, the bottleneck-like methods are inherently excluded from the list. Meanwhile, the explanations for sparsity methods seems to be lack.

2. While the authors provide meaningful and comprehensive plots to demonstrate the results of widening neural networks, the explanations and instructions of the plots and pictures are sometimes missing. For example, In figure 5, the trends of test accurarcy to widening factor are not always positive, some go down when the widening factors are big, others seems to be static. How do they happen? I think further explorations can be made.

3. As for the theoretical side, the authors reference the NTK techniques (Jacot et al., 2018) to explain the relationships between the finite-width neual network and the kernel-based learning. However, NTK techniques are designed for infinite-width neural networks, and have constraints when applied on finite-width networks. As the authors assert, the synchronization of optimizations of 2 methods is only conjecture, which means the prove provide here can only be used to support the NTK theorems. Besides, the relationships between NTK and widening neural networks are not detailedly explained.

II. Quality Part
1. Figure 3, (b) has no horizontal baselines for traditional methods.

---

> ### Author Response · Authors · 2020-11-17
> **Thank you for your thoughtful review! We have updated the paper and addressed all your concerns.**
>
> Thank you for your detailed and thoughtful comments. We are glad that you believe the work is novel and that the experiments are comprehensive. As requested, we added the baseline results to Figure 3(b) in the updated revision. Below, we describe how we have addressed your remaining concern about clarity in the paper revision.
> If you find that your concern is addressed adequately, we would appreciate it if you increase your score to “accept”. Otherwise, if there are any remaining concerns, please let us know.
>
> 1. Re the bottleneck methods:
>
> In the updated version, we have expanded the discussion of the sparsity method and added a figure (Figure 4) that illustrates our sparsification algorithm.
>
> As for the bottleneck methods, we believe it is debatable whether the bottlenecks (and especially the linear bottlenecks) constitute a “big adjustment” -- see for instance AnonReviewer2’s differing opinion on this. Because there is no formal way to quantify all the effects of a structural network transformation, we chose to include the bottleneck methods in our analysis, and to discuss arguments for and against each of our suggested methods.
>
> Including the bottleneck methods leads to a more complete picture of the problem we study, since the bottleneck transformations act at the level of layers, while the sparsity method acts at the level of weights. Apart from this, the bottleneck transformations are common in neural network architectures, and due their practical use we expect that many readers would inquire about the possibility of using these transformations as alternative methods.
>
> 2. Re Figure 5 (Figure 6 in the updated version):
>
> We find generally that performance improves up to some widening factor and deteriorates for yet larger widening factors. The effect is more pronounced for smaller models, and is very mild for the largest models we tested. We discuss this effect toward the end of section 2 in the context of the ImageNet results, and point out (based on additional experiments, shown in Figure 13 in the appendix) that the deteriorating performance appears to be related to a decrease in training accuracy. We have clarified these points in the revised version.
>
> From AnonReviewer2’s comment “The figures in this paper are clear, and each figure was explained in detail. The methods/experimental settings are explained in detail, and the reasons why the authors use these models/methods are also discussed in the paper.” and from our pass over the paper, we believe that Fig. 5 may have been the only instance that needed a bit more explanation. However, if you have noticed other instances of important information missing in or with respect to the figures, please do let us know. Finally, note that we have a dedicated section in the appendix containing additional detailed information about every figure in the main text and the associated experiments.
>
> 3. Re the theoretical analysis:
>
> Thank you for your feedback. We have revised the paper to clarify that the purpose of the theoretical analysis is to explain the main effect observed in the paper for networks with large but finite width. Indeed, the standard NTK results are only valid at infinite width. However, it has been shown that NTK provides a good approximation to network dynamics at large but finite width (see, for example, Lee, Xiao et al. 2019). Also, note that our kernel-based analysis is valid at finite width. This is made possible by the fact that we focus on 2-layer ReLU networks, for which the relevant Gaussian integrals can be performed analytically.

---

### Official Review · AnonReviewer4 · 2020-10-29
**Intriguing discussion which could have been more self-contained**

**Rating:** 6
**Confidence:** 2

**Review:**

This manuscript provides an intriguing discussion on the different roles that the width and parameter size could play in a neural network. While these two aspects are traditionally treated as -if not identical- correlated, the authors managed to develop a couple of configurations to decouple and analyze both separately. Especially the wide and sparse approach could be a new way to design neural networks that are supposed to be small and expressive at the same time.
I'm only concerned about two aspects: i) The section 3 could have been perhaps easier to follow if the work by Jacot et al. 2018 had been briefly recapped. ii) The experiments are all conducted on image dataset. One might wonder whether we could draw the same conclusion for datasets of more general nature.

---

> ### Author Response · Authors · 2020-11-17
> **Thank you for your review! We have addressed your concern in the updated version.**
>
> We thank the reviewer for their helpful comments! As suggested, we have updated section 3 with a review of the Neural Tangent Kernel result.
>
> For your concern (ii) regarding the question whether similar observations could be made on other datasets than images: Indeed, we thought about this question as well. In particular in NLP settings, where the model’s weight count is a major limitation of current experiments, the possible savings that a random static sparsity pattern could provide are of extreme practical relevance. However, extending our study to other learning tasks than image recognition is definitely out of scope for the current paper, and therefore we leave this topic for future work.

---

### Author Response · Authors · 2020-11-17
**Updated version with imrpovements based on all reviews**

We thanks all reviewers for their valuable feedback! We have uploaded a revised version that incorporates improvements based on the reviewers' suggestions.

---

### Decision · Program_Chairs · 2021-01-07
**Final Decision**

**Decision:**

Accept (Poster)

**Comment:**

The paper investigates the interesting question whether increasing the width or the number of parameters is
responsible for improved test accuracy. The paper is very well written and the question is novel and innovative.
From a methodological point of view, the experiments are well conducted, too.
The theoretical part of the paper is somewhat detached from the experimental part and constitutes more of
a heuristic conjecture. In addition, more experiments on a variety of other data sets would have been great.
Ideally, the theoretical section would thus be replaced by such additional experiments, but this is of course not
an option for a conference reviewing system.
Given the innovative question and well-conducted experiments I think that the pros outweighs the cons,
and for this reason I recommend to accept the paper. Reviewer concerns have been well addressed by the authors in their rebuttal and updated version of the paper.